# Transforming Palmyra Atoll to native-tree dominance will increase net carbon storage and reduce dissolved organic carbon reef runoff

**Kate Longley-Wood** [1©]*, **Mary Engels** [2©], **Kevin D. Lafferty**[3‡], **John P. McLaughlin**[4‡], **Alex Wegmann**[5‡]

1 The Nature Conservancy, Protect Oceans Land and Water Program, Boston, Massachusetts, United States of America, 2 Department of Natural Resources and Society, University of Idaho, Moscow, Idaho, United States of America, 3 U.S. Geological Survey, Western Ecological Research Center at Marine Science Institute, University of California, Santa Barbara, Santa Barbara, California, United States of America, 4 University of California, Santa Barbara, Marine Science Institute, Santa Barbara, California, United States of America, 5 The Nature Conservancy, Sacramento, California, United States of America

© These authors contributed equally to this work.
‡ KDL, JPM and AW also contributed equally to this work.
* Kate.Longley-Wood@tnc.org

**Data Availability Statement:** All relevant data are within the paper and its Supporting Information files. Additional spatial datasets are available through the Environmental Data Initiative Data

## Abstract

Native forests on tropical islands have been displaced by non-native species, leading to calls for their transformation. Simultaneously, there is increasing recognition that tropical forests can help sequester carbon that would otherwise enter the atmosphere. However, it is unclear if native forests sequester more or less carbon than human-altered landscapes. At Palmyra Atoll, efforts are underway to transform the rainforest composition from coconut palm (*Cocos nucifera*) dominated to native mixed-species. To better understand how this landscape-level change will alter the atoll's carbon dynamics, we used field sampling, remote sensing, and parameter estimates from the literature to model the total carbon accumulation potential of Palmyra's forest before and after transformation. The model predicted that replacing the *C. nucifera* plantation with native species would reduce aboveground biomass from 692.6 to 433.3 Mg C. However, expansion of the native *Pisonia grandis* and *Heliotropium foertherianum* forest community projected an increase in soil carbon to at least 13,590.8 Mg C, thereby increasing the atoll's overall terrestrial carbon storage potential by 11.6%. Nearshore sites adjacent to *C. nucifera* canopy had a higher dissolved organic carbon (DOC) concentration (110.0 μMC) than sites adjacent to native forest (81.5 μMC), suggesting that, in conjunction with an increase in terrestrial carbon storage, replacing *C. nucifera* with native forest will reduce the DOC exported from the forest into in nearshore marine habitats. Lower DOC levels have potential benefits for corals and coral dependent communities. For tropical islands like Palmyra, reverting from *C. nucifera* dominance to native tree dominance could buffer projected climate change impacts by increasing carbon storage and reducing coral disease.

Portal, a publicly-accessible data repository. Palmyra Atoll dissolved organic carbon sampling locations and values: doi:10.6073/pasta/1a257081daf08adb5eb665a192cd370a; Locations of rainforest transformation plots on Palmyra Atoll: doi:10.6073/pasta/71aaeac403060543f85ca1452665b56e; Palmyra Atoll soil and/or wood density sampling locations used in the carbon storage analysis: doi:10.6073/pasta/d30d7d79c4357bf35973b69932151344.

**Funding:** This project was made possible with funding from the Wildlife Conservation Society Climate Adaptation Fund – established by a grant from the Doris Duke Charitable Foundation (received by AW in 2018 – no grant number provided) and a grant from the National Science Foundation. Additional data collection was funded by the NSF DEB #1457371. The funders had no role in study design, data collection and analysis, decision to publish, or preparation of the manuscript.

**Competing interests:** The authors have declared that no competing interests exist.

## Introduction

Planting trees can be a win-win for nature when the land to be restored is unforested, but proposals to replace non-native trees like coconut palms (*Cocos nucifera*) with native forest habitat can lead to societal conflicts and could impact carbon storage and cycling. At Palmyra Atoll ("Palmyra"), efforts are underway to transform the rainforest community composition from *C. nucifera* dominated to native mixed-species. Here, we use the term transform, rather than restore because the management objective is not to recreate a predetermined baseline vegetation community, but rather to optimize community conditions to enhance ecological outcomes. Palmyra rainforest management and transformation efforts have conservation benefits without societal conflicts; however, it is less clear how changes in forest composition will alter Palmyra's carbon stocks and flows. Although relatively isolated from human disturbance, Palmyra, like other island ecosystems [1, 2], is impacted by invasive plants [3]. *C. nucifera* has historically been recorded as native vegetation [4]; however, it is now naturalized outside of the species' southeast Asian place of origin, and is characterized as introduced or non-native to Palmyra in the existing literature [3, 5–7]. Historical cultivation for copra at Palmyra beginning in the 19th century [4, 5, 8] and subsequent natural range expansion led to *C. nucifera* occupying ~40% of the forest canopy today [9].

The *C. nucifera* expansion came at the expense of Palmyra's formerly large native forests. The ten tree species native to Palmyra compete with *C. nucifera* for light [10] and freshwater [5, 11], and losses to native forests have indirect impacts on ecosystem dynamics of Palmyra's rainforest. Less food for invertebrate herbivores [12] under the *C. nucifera* canopy has reduced prey for native geckos [3], the only resident terrestrial vertebrate. Moreover, nitrogen and phosphorus deposits are orders of magnitude lower under *C. nucifera* than under the native broadleaf canopy preferred by roosting seabirds [3, 13]. In marine environments, the proximity of nutrient-poor palm-dominated forests has been linked to a decrease in plankton abundance and growth, impacting the foraging behavior of manta rays [3, 14]. The loss of native forest cover, especially the decline of the iconic atoll native *Pisonia grandis* [12], coupled with the expansion of *C. nucifera*, motivated efforts to transform Palmyra's rainforest from a palm-dominated canopy to a mixed-species rainforest with native trees and shrubs.

Spearheaded by the United States Fish and Wildlife Service (USFWS) and The Nature Conservancy (TNC), these management efforts are expected to take five years and include removal of *C. nucifera* from 95% of the emergent land area and replanting native forest species. The forest transformation program at Palmyra started in 2019 [41]. Forests fix carbon through photosynthesis, and forest biomass (i.e., roots, trunks, branches, and leaves) is approximately 50% carbon [15]. Some of this carbon-rich biomass eventually falls to the forest floor as litter, which then decomposes and adds to soil organic carbon (SOC) [16]. The amount of carbon stored in forest biomass and soils is largely influenced by climatic conditions [15]; however, tree species can alter carbon budgets at the local scale [17–19]. With rain, the water soluble fraction of SOC becomes dissolved organic carbon (DOC) [20], which can leach into adjacent water bodies. The DOC flux in temperate forest is between 6% and 30% (average 17%) of aboveground litter input [21, 22]. DOC affects various processes on coral reefs, including favoring bacterial populations associated with coral disease [23]. The rate of DOC flux varies by vegetation type [21], and increases with precipitation, pH, and decomposition [22], environmental conditions that are common at Palmyra, though fine scale data on these metrics do not exist at the scale of the sampling site. This study aims to assess the carbon implications of the ongoing forest management and transformation efforts at Palmyra with the hypothesis that an increase in the ratio of native to non-native trees will have a net positive increase on carbon accumulation–an added benefit of planned management activities. To project carbon-

related impacts, this study measured existing carbon budgets in Palmyra's *C. nucifera*-dominated forest and remnant native mixed-species forest. We used field sampling, remote sensing, and parameter estimates from the literature to evaluate above-ground carbon storage, SOC accumulation and DOC export. This data informed models of how the atoll's carbon budget and export will change under *C. nucifera* removal and replacement with native trees and shrubs. *C. nucifera* is widespread throughout Oceania [7], and the results from this study can inform efforts to conserve and restore native-tree dominance on other islands.

## Methods

### Estimating soil carbon

**Soil carbon field sampling.** Field sampling was conducted within the Palmyra Atoll National Wildlife Refuge (5˚52' N, 162˚04' W), jointly managed by TNC and USFWS. Research reported here was conducted under USFWS Special Use Permits 12533–19013 and 12533–16007. No protected species were sampled.

To estimate soil carbon storage across Palmyra's dominant tree species, we structured our sampling to account for how islet size and vegetation community, (and by extension, nutrient subsidies) could influence soils. The 17 islets (out of a total of 28) sampled in this study form a representative subset in terms of relative size: 8 large (>6.5 ha), 5 medium (between 1 and 6.5 ha) and 4 small (<1 ha). In 2016, 72 sampling sites spanning seven canopy types were chosen at random after stratifying by islet and canopy type. Sampling efforts focused on the most widespread and dominant canopy types: *C. nucifera* (N = 15), *Pandanus tectorius* (N = 15), *Scaevola sericea/Heliotropium foertherianum* (N = 15) and *P. grandis* (N = 14). Canopy types with more restricted distributions were sampled less: *Terminalia catappa* (N = 1), no canopy cover (N = 9), and *Hibiscus tiliaceus* (N = 3). The replicates for each canopy type came from different islets, except *H. tiliaceus* which had multiple sites from one islet due to its limited distribution.

We augmented the 2016 sampling data in 2019 by adding 27 sites (totaling 99 sites), evening the distribution between samples taken in homogeneous tree communities (n = 47) and mixed-species communities (n = 52). This was beneficial as mixed tree stands can have different carbon accumulation characteristics than pure stands [24]. This also increased the number of samples collected from islets on the western edge of the lagoon, where dredging activities during the mid-20th century resulted in man-made islets (Sand, Lesley, and Dudley). These islets may have different soil properties and thus different carbon storage capacities than soils occurring on natural islets. Samples from both time periods were assigned a vegetation community type based on remote sensing data [9]. 2019 sampling efforts covered the five most dominant woody-stemmed species (*C. nucifera*, *H. foertherianum*, *P. tectorius*, *P. grandis*, and *S. sericea*) on five islets (Cooper, Eastern, Dudley, Sand, and Lesley Islets). Sampling locations are shown in Fig 1. Table 1 lists the number of samples by islet and islet area and Table 2 lists the number of samples by vegetation community type, along with the proportional representation of each community type in the study area. Additional sample data can be found in S1 Table.

At each sampling location, 1,000–2,000 ml of soil were removed from the first 0–15 cm of soil with a small trowel and stored in a sealable plastic bag. Because the coarse textured nature of atoll soils makes traditional coring difficult, we followed the water method for sample volume [25], which determines volume by lining the sample hole with a thin plastic layer and recording the volume of water needed to fill the hole to the reference surface. Hardpan depth at each site was measured with a soil probe. After collection, soil samples were passed through

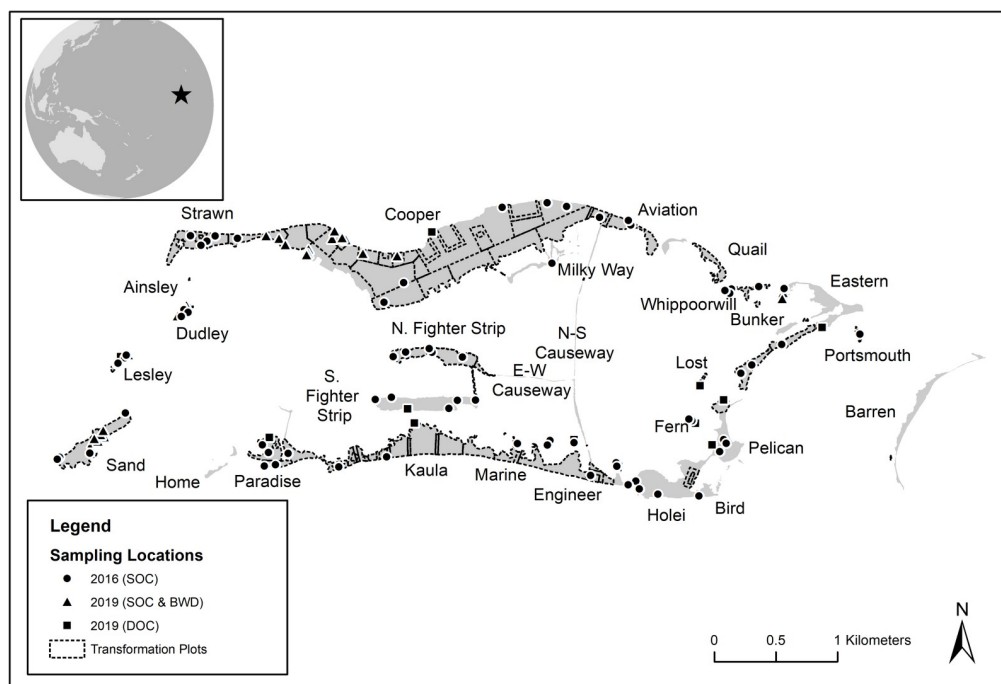

**Fig 1. Map of study area at Palmyra.** Sampling locations and years are provided for soil organic carbon (SOC), basic wood density (BWD) and dissolved organic carbon (DOC). Transformation plot boundaries are also depicted. Basemap data depicting Palmyra Atoll islet locations are made available in the public domain from the USGS [9]. Global reference map in the inset is made available in the public domain from Natural Earth.

**Table 1. Soil samples by islet.**

| Islet | No. Samples | Islet Area (ha) |
|---|---|---|
| Aviation | 3 | 4.28 |
| Cooper | 18 | 91.81 |
| Dudley | 7 | 0.89 |
| Eastern | 7 | 11.43 |
| Engineer | 7 | 6.52 |
| Fern | 2 | 0.25 |
| Holei | 4 | 11.42 |
| Kaula | 5 | 14.18 |
| Lesley | 6 | 0.61 |
| N. Fighter | 5 | 6.13 |
| Pelican | 3 | 6.07 |
| Paradise | 5 | 6.21 |
| Portsmouth | 1 | 0.26 |
| S. Fighter | 5 | 8.12 |
| Sand | 11 | 6.92 |
| Strawn | 5 | 7.04 |
| Whippoorwill | 5 | 1.18 |

Number of soil samples by islet, with the area of each islet.

**Table 2. Soil samples by community.**

| Vegetation community | # of Samples | % Vegetated Area |
|---|---|---|
| *C. nucifera* | 20 | 29.8% |
| *C. nucifera/H. foertherianum* | 16 | 11.5% |
| *H. foertherianum/S. sericea* | 3 | 4.0% |
| *H. tiliaceus* | 2 | 2.6% |
| *Lepturus repens var. palmyrensis/Fimbristylis cymosa* | 2 | 6.5% |
| *P. tectorius* | 12 | 13.0% |
| *P. grandis* | 29 | 13.2% |
| *P. grandis/H. foertherianum* | 14 | 1.4% |
| *S. sericea/H. foertherianum* | 1 | 11.8% |
| *Other* | 0 | 6.3% |

Number of soil samples by vegetation community, and proportional representation of each community type across Palmyra's vegetated area.

a 2 mm sieve, air-dried, and then the <2 mm fraction was shipped to a laboratory for carbon analysis.

The samples collected in 2016 were analyzed for both organic carbon and total carbon percentage. Although the 2019 samples were analyzed for total carbon only, the organic carbon fraction for the 2019 samples were estimated based on the relationship between total carbon and organic carbon derived from the 2016 samples and samples collected at other atolls in the tropical central Pacific.

**Soil carbon modelling approach.** To estimate soil carbon values across the study area we used the 'randomForest' statistical package in R [26, 27] to predict soil carbon percentages using 1 m resolution remotely-sensed vegetation community and tree species dataset as predictor variables. These variables were selected due to the high-resolution spatial data coverage across the study area. The "Woody Crowns Palmyra Atoll 2016" dataset [9] from which the tree species raster was derived contained overlapping polygons, and each pixel in the resulting raster was assigned the tree species value that was most prevalent in the 1 m cell. Similarly, the vegetation community raster was derived by converting the "Vegetation Communities Palmyra Atoll 2016" polygons [9] to a 1 m raster. To account for known soil heterogeneity [28], soil carbon percentages for each sample were binned as high (>20%), medium (10–20%), and low (<10%), informed by the Natural Resource Conservation Service organic/mineral soil definitions [29] before running the model. Total carbon was calculated by multiplying the soil percentage value associated with each category by the mean dry bulk density for each category (S1 Table), the soil depth (assumed to be 20 cm based on average hardpan depth measurements across all soil sampling locations), and total area.

To verify the results of the random forest model, we also calculated present-day soil carbon using the mean carbon concentration and dry bulk density values based on the combination of vegetation community and species at each location. To do so, we converted the 1m community and tree species rasters to polygons and used a geometric union to create a polygon layer identifying community type and tree species across the study area. We then assigned mean carbon concentrations and dry bulk density values to each polygon by calculating averages for each combination of variables (vegetation community and species) in in the sampling data. Where a combined mean could not be calculated from the data (i.e., no values for a specific species was measured or when n < 2 for a combination of community and species), the community mean value was used as a default. Similarly, if there was no value for the community, the

species value was used. If neither value was available, the mean of all samples was used. We assigned a value of zero to all areas where either community or species value was blank, as this usually indicated a shoreline polygon overlapping with water, or where either value was either unknown, water, or the runway area on Cooper Islet. We then calculated present day carbon values using the same formula described above.

The presence of man-made islets as part of the atoll system provides a unique opportunity to estimate soil carbon accumulation rates for Palmyra's vegetation communities. Dudley and Leslie Islets' primary vegetation community is *P. grandis*/*H. foertherianum*, and the construction of these islets was completed by 1942. Similarly, the North Fighter Strip Islet, home to communities of *H. foertherianum*/*C. nucifera* was likely completed by 1943 [30]. We assumed a linear accumulation rate and estimated the yearly soil accumulation rate for the *P. grandis*/*H. foertherianum* and *H. foertherianum*/*C. nucifera* communities by dividing the current estimated soil carbon totals on these islets and dividing them by 77 years and 76 years, respectively, using 2019 as the baseline year to indicate the minimum accumulation rate. As there are no homogeneous *C. nucifera* communities on the man-made islets sampled in this study, an accumulation rate was estimated from available literature [31] by dividing the reported carbon density at 0–20 cm sampling depth (27 Mg C ha$^{-1}$) by 20 (the age of the trees). By multiplying the carbon accumulation rates by the current and projected area estimates for each community type and then by 15 for the number of years until projected community maturity, we were able to compare the projected carbon accumulation over 15 years in a no-transformation scenario to our current transformation scenario, to determine the additional carbon added under the transformation scenario.

## Estimating aboveground carbon

**Aboveground carbon field sampling.**   We derived our basic wood density (BWD; i.e., the ratio of dry mass over green volume) estimates from the literature and field sampling. BWD estimates for *C. nucifera*, *Cordia subcordata*, *H. tiliaceus*, *and T. catappa* were from the French agricultural research and international cooperation organization (CIRAD) database [32]. BWD estimates for many Palmyra native trees are not present in these databases, so we collected wood from *P. grandis*, *H. foertherianum*, *P. tectorius*, and *S. sericea* at Palmyra. Wood was collected at 18 different locations. As the location at Palmyra was not expected to influence variation of wood density, the sites were not stratified by islet, and instead were collected opportunistically to coincide with soil sampling locations (Fig 1). At each sampling location, a single woody branch approximately 10 cm in length was removed from each tree and cut into discs.

To standardize sampling in a remote location with limited facilities, woody density was measured (disc diameter and width) at the fiber saturation point [33] (discs were submerged for 24 hours at ambient pressure) and the anhydrous state (measured and weighed). We did not have the ability to estimate fiber saturation point and volumetric shrinkage in the field. These values were estimated as the mean of values reported for 50 tropical tree species by the Inter-American Institute for Cooperation on Agriculture (IICA) [34], with the exception of *P. grandis*, for which genera-specific values were reported. The ranges for both values were narrow and final BWD estimates were insensitive to variation within reported ranges. With these coefficients, we calculated BWD for native trees at Palmyra according to Vieilledent et al. [32], making them comparable to CIRAD values. BWD values for Palmyra natives fell within the range reported for congeners in CIRAD [32]. The corrected BWD values (i.e. the minimum, mean, and maximum values that were calculated using these coefficients) and their sources can be found in Table 3.

**Table 3. Corrected basic wood density (BWD) values by species.**

| Species | BWD Min (g cm-3) | BWD Max (g cm-3) | BWD Mean (g cm-3) | Source(s) | SE | Sample Size (n) |
|---|---|---|---|---|---|---|
| *Cocos nucifera* | 0.31 | 0.70 | 0.50 | CIRAD[a], Reyes et al.[b] | 0.08 | 5 |
| *Cordia subcordata* | 0.18 | 0.89 | 0.42 | CIRAD, Reyes et al., Hidayat & Simpson[c], GWDD[d] | 0.01 | 135 |
| *Heliotropium foertherianum* | 0.23 | 0.44 | 0.31 | Field data, GWDD | 0.01 | 31 (15) |
| *Hibiscus tiliaceus* | 0.26 | 0.66 | 0.38 | CIRAD, GWDD | 0.02 | 28 |
| *Pandanus tectorius* | 0.06 | 0.32 | 0.16 | Field data, GWDD | 0.01 | 43 (21) |
| *Pisonia grandis* | 0.12 | 0.46 | 0.23 | Field data, GWDD, Hidayat & Simpson, IICA[e], Reyes et al. | 0.01 | 51 (21) |
| *Scaevola sericea* | 0.23 | 0.57 | 0.40 | Field data | 0.06 | 6 (3) |
| *Terminalia catappa* | 0.22 | 0.83 | 0.47 | CIRAD, GWDD, IICA, Reyes et al. | 0.01 | 330 |

The source notes where the values used to calculate the corrected BWD came from (either literature, field data, or both). For values calculated from field measurements collected during this study, both the raw and corrected values were used to calculate the mean. The sample size refers to the number of calculations used in the correction while the number in parentheses indicates the number of samples collected in the field.

[a]CIRAD [32].

[b]Reyes et al. [35].

[c]Hidayat & Simpson [36].

[d]GWDD (Global Wood Density Database) [37].

[e]IICA [34].

**Aboveground carbon modelling approach.** Due to lack of data availability on key metrics, we were only able to estimate biomass values for the dominant tree species at Palmyra. The list of species used in this calculation are shown in Table 3.

We estimated aboveground biomass (AGB) using the equation derived specifically for tropical tree species [38], that incorporates (BWD) ($\rho$), tree height ($H$), and diameter ($D$) where $AGB = .0673 \, X \, (\rho D^2 H)^{.976}$. The mean corrected BWD values (Table 3) were used in this equation.

The location, species, and height of each tree was derived from remote sensing data [9, 39]. To approximate the point location of each tree trunk, we used the remote sensing dataset representing woody crowns to derive the centroid of each crown polygon. In order to estimate the approximate height of each tree, we then used the elevation data to assign a height to each point, subtracting 2m from each location to account for height above sea level. A gap in the LIDAR data [39] for Engineer Islet resulted in some of the points being assigned a mean height value specific to the species across the study area. Diameters were assigned according to tree species by taking the quadratic mean of measured tree diameters collected during transect surveys (S3 Table). Aboveground biomass values were then calculated using the above-referenced equation and then converted to carbon values using a multiplier of 0.47 [40]. The carbon density (Mg C ha$^{-1}$) is based on an estimate of areas covered by a canopy (i.e. area that excludes runway, bare ground, grassland, and unknown land cover).

## Projecting future soil and aboveground carbon

To estimate the expected carbon storage following transformation activities (allowing for transformed forests to reach maturity), we created spatial data layers for expected future vegetation community values and future tree species values, based on a transformation area polygon dataset provided by Island Conservation [41]. Within the transformation polygons, the objective is to remove all *C. nucifera*, and plant *H. foertherianum* and *P. grandis* at a 2:1 ratio.

However, since the survival rate of replanted *H. foertherianum* is estimated at 46% (S4 Table), mature, transformed forest communities are expected to have approximately half of the existing *C. nucifera* replaced by *H. foertherianum* and half replaced by *P. grandis* [41]. These forest communities will be considered mature 15 years after they have been planted.

Based on the management plan [8, 41], we simulated the future community scenario data by selecting areas within the remote sensing data [9] that had a community value of *C. nucifera* and were within the transformation polygons and reclassified them as the mixed *P. grandis*/*H. foertherianum* community. Similarly, all tree species points classified as *C. nucifera* within the transformation areas were randomly reassigned to be either *P. grandis*, or *H. foertherianum*, based on the target 50:50 representation. Heights for newly planted *P. grandis* and *H. foertherianum* were assigned based on mean values for those species across the study area. Using the steps described in previous sections, we then recalculated the aboveground and soil carbon values based on predicted future community and species values.

### Dissolved organic carbon

DOC was sampled at 12 sites (Fig 1) to compare DOC export in native and non-native canopies, though this sample size was deemed too small to try to model DOC concentrations under the future transformation scenario. Sites were chosen to represent different parts of the atoll, shorelines facing different directions (N, S, E, W), different canopy types and different C. nucifera removal schedules. Because metabolism by marine organisms can alter DOC flux, and respiration and photosynthesis are affected by temperature and light, three HOBO Pendant® Temperature/Light Data Loggers were place on the shoreline for 24 hours at each DOC site, and at least one logger returned air temperature and lux each minute for each site (we report the average daily value per site across all functioning loggers). Lux was converted to relative lux by dividing values at a site by values at an unshaded site. To confirm that shading reduced water temperature in the nearshore, we compared four submerged shaded loggers with two submerged unshaded loggers.

Two samples were taken 50 m apart at each site. Three comparison samples (expected to be low in terrestrial influence) were taken from surface waters offshore of the atoll, on the forereef and from lagoon away from shore. Water samples were collected in new quart Ziploc bags with powder free gloves and put on ice after collection. To process DOC, Luer lock syringes were filled with 5% HCl for > 1 hour, then rinsed 2 more times with acid, then rinsed with distilled water before use. Filter holders were similarly soaked in 5% HCl and rinsed. The DOC filtering procedure involved drawing 60 ml of water from the sample bag into a syringe as a rinse and then then drawing a second 60 ml for the sample. A combusted GF75 filter was loaded into a clean 25 mm plastic filter holder and connected to the syringe. The syringe was used to filter 30 ml of sample to rinse an acid-washed collection vial three times. The last 30ml was inserted into the vial and the vial was capped. One filter was used per site. Filtered samples were acidified to pH 2- by adding 50ul 4N HCl, then agitated to mix. Samples were stored and shipped to Dr. Craig Carlson's lab at the University of California, Santa Barbara for analysis. Samples were analyzed via high temperature combustion method on a modified Shimadzu TOC-V or Shimadzu TOC-L using the standardization and referencing approaches described in [42]; detection limit was roughly 1 μmol C L$^{-1}$.

## Results

### Soil carbon

The soil organic carbon percentages calculated from sample data ranged from 0.48–38.46%, with the mean value of 6.37%. There were 83 samples in the Low bin, 9 in the Medium bin,

**Table 4. Summary of soil organic percentage values derived from sample data.**

| Bin | No. Samples | Organic Carbon Percentage Range | Organic Carbon Percentage Mean | Organic Carbon Percentage SE | Dry Bulk Density Range | Dry Bulk Density Mean | Dry Bulk Density SE |
|---|---|---|---|---|---|---|---|
| Low | 83 | 0.48–9.93 | 3.64 | 0.25 | 0.15–2.09 | 1 | 0.05 |
| Medium | 9 | 10.16–15.95 | 12.88 | 0.69 | 1.17–3.84 | 0.43 | 0.11 |
| High | 7 | 21.80–38.46 | 30.43 | 2.7 | 0.10–0.54 | 0.27 | 0.06 |

Organic carbon percentage values by bin. Carbon percentage and dry bulk density value ranges, means, and standard errors are also reported.

and 7 in the High bin (Table 4). The values calculated for each sample, the islet where they were collected, and the assigned bin can be found in S1 Table.

Based on these values, the random forest analyses predicted soil carbon storage values comparable to the values calculated using averages from vegetation communities and species measurements. The latter approach predicted baseline values that were comparable (11,856.2 Mg C in the averages method compared to 11,872.2 Mg C in the random forest method) and future values that were somewhat higher than those predicted by the random forest analysis (15,378.2 Mg C in the averages method compared to 13,590.8 in the random forest method). For this reason, only the results of the random forest model are presented, as they reflect a more conservative estimate for changes in carbon storage.

The results of the soil carbon analyses indicate a 14.5% gain in soil carbon in the post-transformation scenario, representing an increase from 73.4 Mg C ha$^{-1}$ to 84.0 Mg C ha$^{-1}$ when averaged across the study area (Table 5). The model projects a reduction in soil carbon contributions from *C. nucifera*, *H. tiliaceus*, and *S. sericea*, and an increase in soil carbon contributions from *H. foertherianum*, *P. tectorius*, *P. grandis*, and *T. catappa* in under the transformation scenario (Fig 2A). Kaula Islet (14.2 ha) shows the largest gain, by total tonnage, with an increase of 509.9 Mg C, and the second largest gain by percentage (57.0%)—Cooper Islet (91.8 ha) shows the second largest gain by soil carbon, with an increase of 347.0 Mg C (6.9%). Marine Islet (5.5 ha) has the largest gain by percentage, with a percent increase of 63.7% (217.7 Mg C) (S5 Table). A map of the distribution of soil carbon in the pre and post transformation scenarios and areas of loss and gain can be found in Fig 3A–3C.

Estimates of current carbon soil content in the *P. grandis/H. foertherianum* and *C. nucifera/H. foertherianum* communities on manmade Dudley, Lesley, and North Fighter Strip Islets suggest a soil carbon accumulation rate of 1.3 Mg Carbon ha$^{-1}$ year$^{-1}$ for *P. grandis/H. foertherianum* and 1.0 Mg C ha$^{-1}$ yr$^{-1}$ for *C. nucifera/H. foertherianum*. 1.3 was the estimated accumulation rate for *C. nucifera* derived from the literature [31]. This suggests that after 15 years, there will be marginally less carbon accumulation in the transformation scenario (1,517.3 Mg C) compared to the no transformation scenario (1,536.9 Mg C) (Table 6).

**Table 5. Summary of current and projected total carbon.**

| Carbon source | Baseline Total (Mg C) | Baseline Density (Mg C ha$^{-1}$) | Future Total (Mg C) | Future Density (Mg C ha$^{-1}$) | % Change to Total |
|---|---|---|---|---|---|
| Aboveground | 692.6 | 4.3 | 433.3 | 2.7 | -37.4% |
| Soil | 11,872.3 | 73.4 | 13,590.8 | 84.0 | +14.5% |
| **Total** | **12,564.9** | **77.6** | **14024.1** | **86.7** | **+11.6%** |

The aboveground carbon values reported are calculated based on the mean basic wood density values. The carbon density (Mg C ha$^{-1}$) is based on an estimate of areas partially or fully covered by a canopy (i.e. all areas other than those classified as runway, bare ground, grassland, and unknown land cover).

## Basic wood density

Corrected basic wood density values for the dominant woody vegetation types at Palmyra are reported in Table 3. The uncorrected BWD values calculated from field data ranged from 0.16 (*P. tectorius*) to 0.40 (*S. sericea*). These raw, pre-corrected BWD values can be found in S2 Table. S1 Fig compares BWD estimates from the literature to both uncorrected and corrected values from field data.

## Aboveground carbon

The results of the analyses indicate a decline of 37.4% in total aboveground carbon in the post-transformation scenario, representing a decrease from 4.3 to 2.7 Mg C ha$^{-1}$ when averaged across the study site (Table 5). On average, *C. nucifera* at Palmyra exceed both *P. grandis* and *H. foertherianum* in height, density, and basic wood density, meaning that their removal and replacement results in a loss of aboveground biomass, and, by extension, aboveground carbon stocks. In the post-transformation scenario, the gain in aboveground carbon storage from the addition of *P. grandis* and *H. foertherianum* does not offset the simultaneous losses in carbon storage from the removal of *C. nucifera* (Fig 2B).

Kaula Islet shows the largest loss, both by percentage and total tonnage, with a decrease of 73 Mg C (69%). Cooper Islet shows the second largest loss by total carbon, with a decrease of 65 Mg C. Aviation Islet (4.3 ha) has the second largest loss by percentage, with a percent decrease of 65% (S5 Table). This is similar to the spatial patterns observed for soil organic carbon increases. A map of the distribution of aboveground carbon in the pre and post transformation scenarios and areas of loss and gain can be found in Fig 3D–3F.

## Overall carbon

Overall, the gains in soil carbon in the post-transformation scenario offset the losses to aboveground carbon, resulting in an overall increase (11.6%) in carbon storage in the post-transformation, representing an increase from 77.6 Mg C ha$^{-1}$ to 86.7 Mg C ha$^{-1}$ when averaged across the study area (Table 5) after transformation maturation at 15 years. The large gains in soil carbon storage driven by the increased number of *P. grandis* and *H. foertherianum* eclipse the loss of aboveground carbon driven by the removal of *C. nucifera* (Fig 2C). A map of the distribution of total carbon in the pre and post transformation scenarios and areas of loss and gain can be found in Fig 3G–3I.

## Dissolved organic carbon

Intertidal (81.5 μMC) sites that were adjacent to native forest did not differ in DOC from offshore (71.9 μMC), forereef (78.9 μMC), or open water lagoon (73.8 μMC) sites. Water depth was not a significant factor after accounting for habitat type. If sites were adjacent to *C. nucifera* there was no difference (p = 0.9) in their DOC concentration whether they came from the intertidal lagoon flats (117.0 μMC) or intertidal reef flats (112.7 μMC). We pooled all the intertidal samples adjacent to *C. nucifera* canopy before comparing canopy effects with site as random effect to account for repeated measures at a site. We also excluded Sacia Islet because *C. nucifera* had recently been controlled there. Intertidal sites adjacent to *C. nucifera* canopy had a higher DOC concentration (110.0 μMC) than intertidal sites adjacent to native forest (81.5 μMC, P = 0.0025, Fig 4). Note that Saci1E, where *C. nucifera* had been controlled, had a lower DOC (69.1 μMC), than any site with intact *C. nucifera*, suggesting that the hypothesized *C. nucifera* effect could be rapidly reduced after *C. nucifera* removal (Fig 4). Air temperature declined with shade, and shading reduce afternoon water temperatures from 37 to 34 degrees,

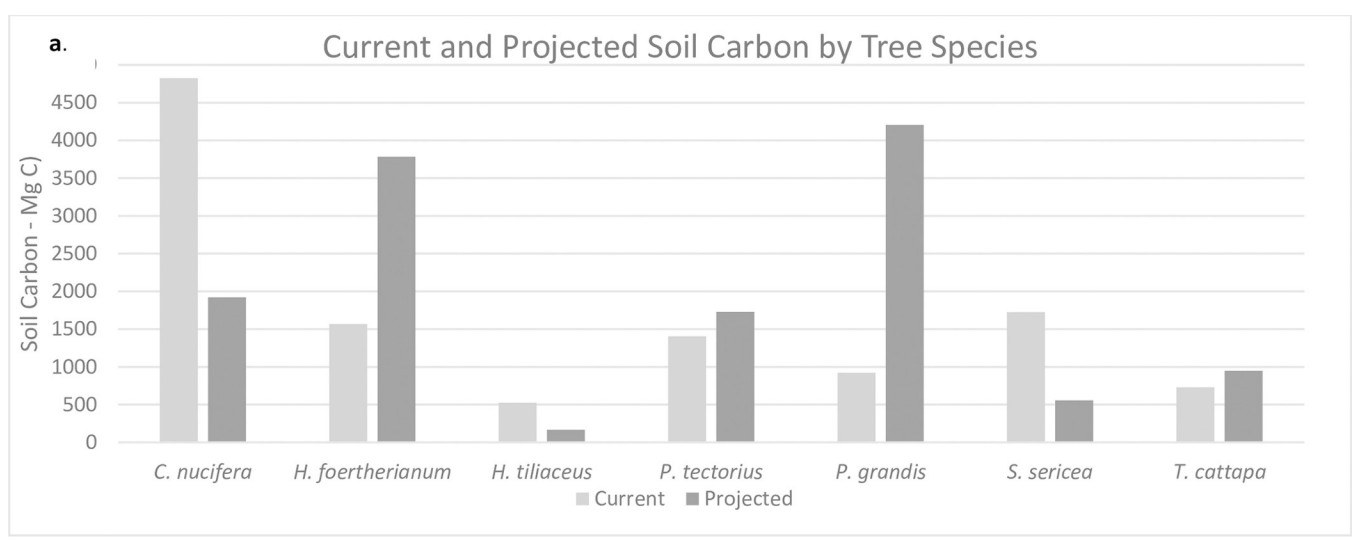

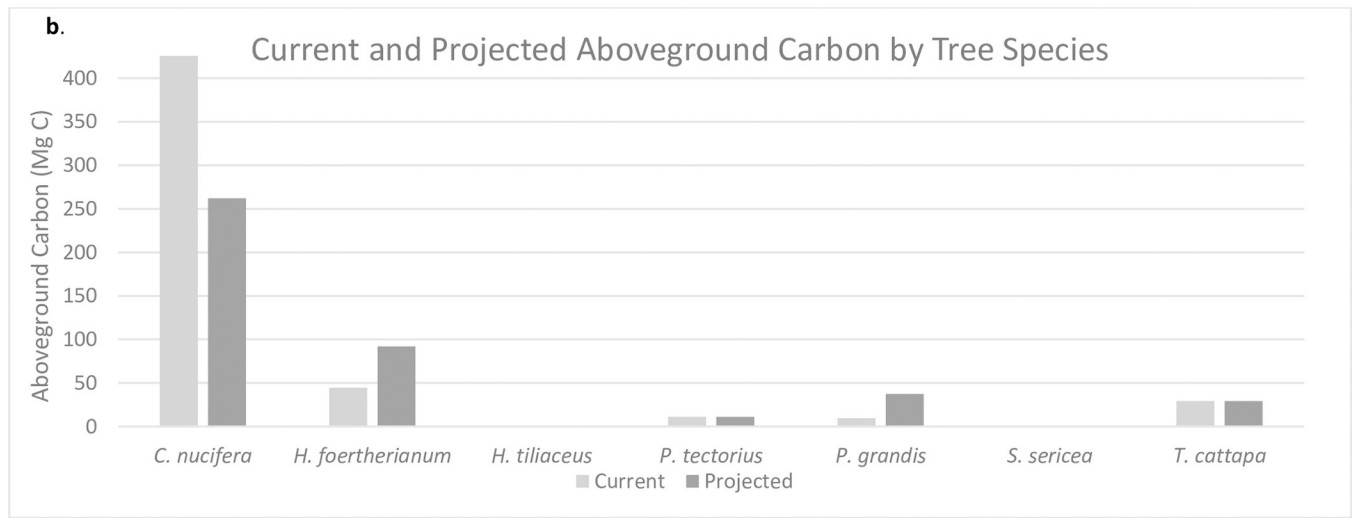

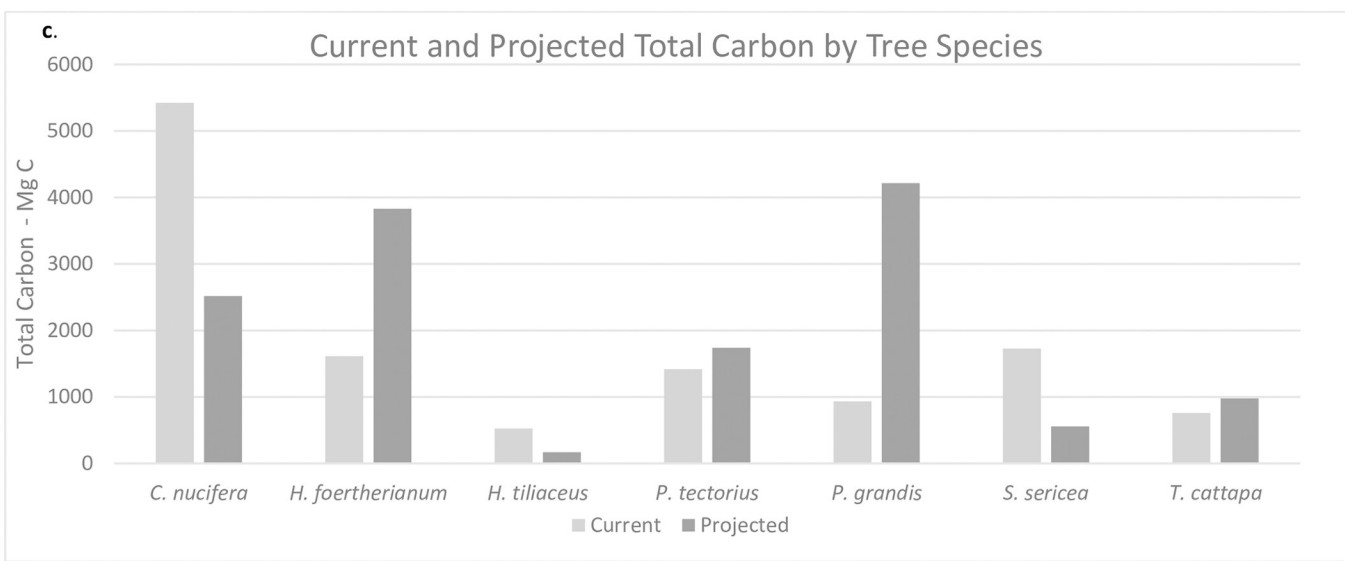

**Fig 2.** Current and future projections of soil organic carbon, (2a), aboveground carbon (2b), and total carbon (soil and aboveground) (2c) by tree species.

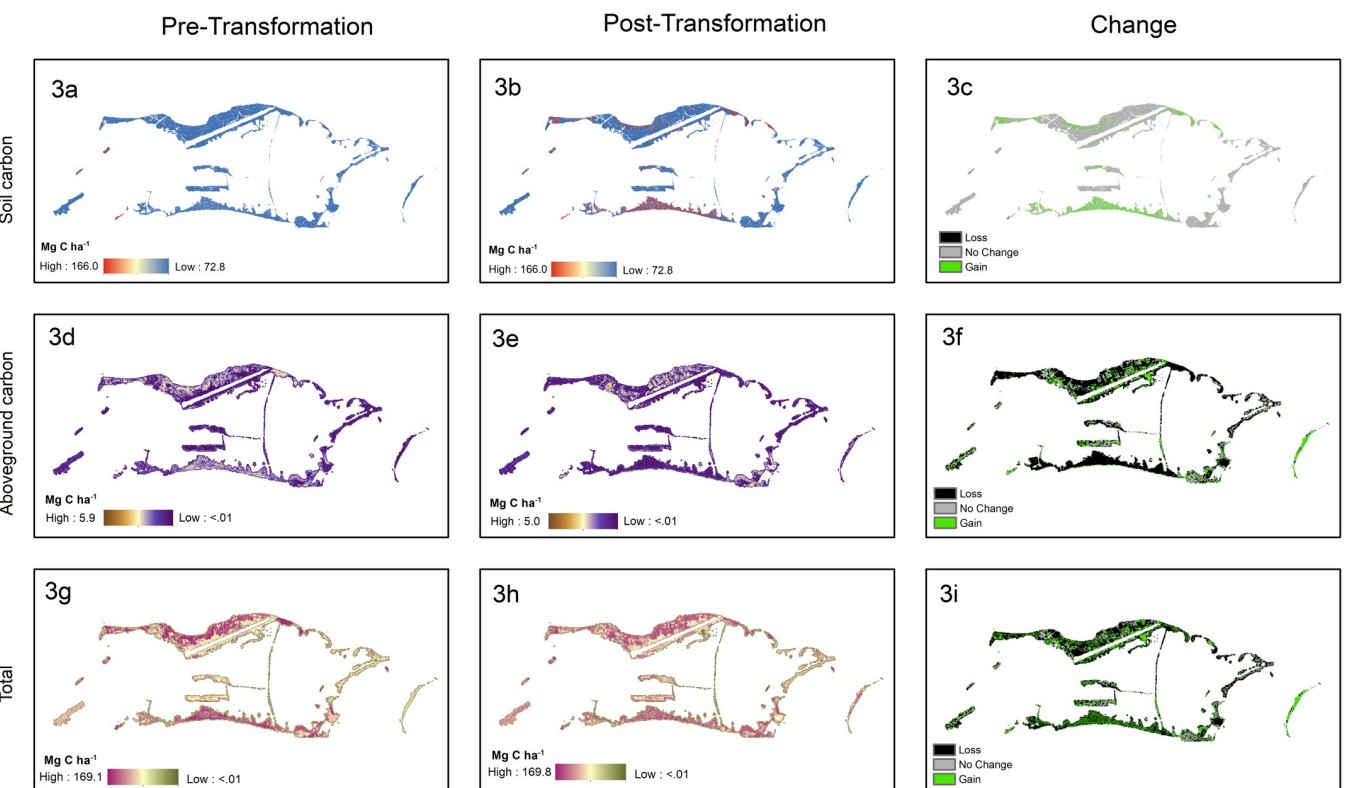

**Fig 3. Maps of carbon distribution across the study area in the pre (column 1) and post (column 2) transformation scenarios.** The third column shows the net gain and loss of carbon between the two scenarios. Fig 3A–3C show the results for soil carbon. Fig 3D–3F show the results for aboveground carbon. Fig 3G–3I show the sum of the two carbon sources. Units are in Mg C ha$^{-1}$ where the area includes only those areas partially or fully covered by a canopy (i.e. all areas other than those classified as runway, bare ground, grassland, and unknown land cover).

but neither the average 24-hour air temperatures (mean = 32.6 C, SD = 1.7), nor relative Lux values (0.41,SD = 0.20), taken at DOC sites differed by canopy type, or correlated with DOC, suggesting that the differences among sites was not greatly affected by differences in metabolism from marine organisms.

A conceptual illustration of study results and other anticipated transformation and realignment benefits can be found in Fig 5.

## Discussion

This study is the first known attempt to calculate the carbon stored by vegetation for an entire tropical atoll and represents a unique opportunity to explore the benefits of transformation

**Table 6. Calculations of soil carbon accumulation estimates in transformation and no transformation scenarios based on changes to community structure.**

| Community | Total ha (Current) | Accumulation Rate (Mg C ha$^{-1}$ yr$^{-1}$) | Total accumulation after 15 years with no transformation (Mg C) | Total ha (projected) | Total accumulation after 15 years with transformation (Mg C) |
|---|---|---|---|---|---|
| *C. nucifera* | 57.3 | 1.3 | 1160.5 | 14.5 | 292.7 |
| *H.foertherianum/P. grandis* | 2.7 | 1.3 | 51.1 | 54.0 | 1020.3 |
| *H.foertherianum/C. nucifera* | 22.6 | 1.0 | 325.3 | 14.2 | 204.3 |
| **Total** | | | **1536.9** | | **1517.3** |

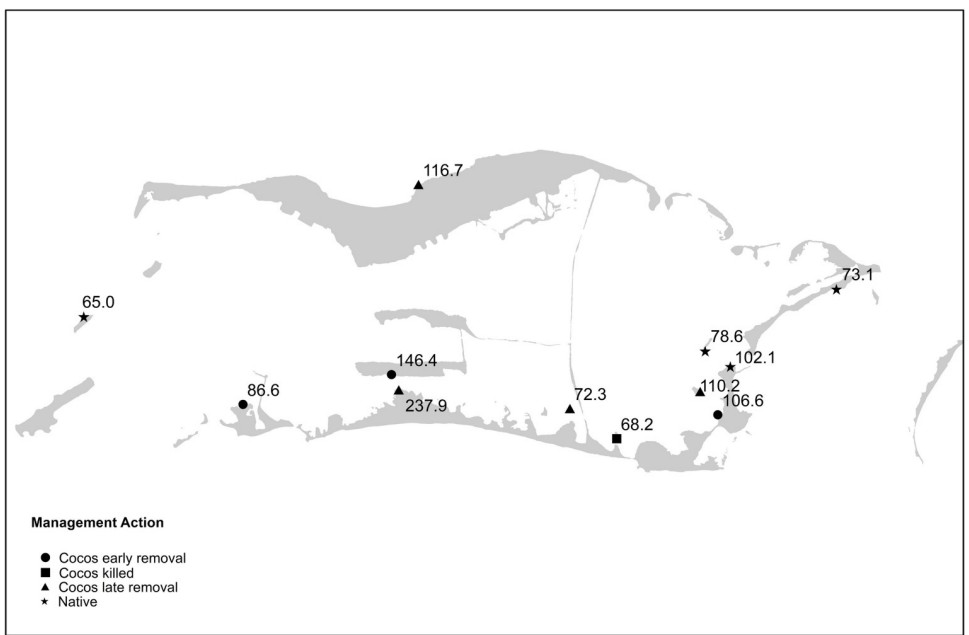

**Fig 4. Map of DOC sampling results.** On this map, different shapes symbolize different management actions under which the samples were taken; DOC, (µMC) measured at each site is indicated by a label. Basemap data depicting Palmyra Atoll islet locations is made available in the public domain from the USGS [9].

beyond biodiversity and ecosystem health in a manner that can inform management activities in other similar environments. The results of the study confirmed that *C. nucifera* and native forest types differed in their carbon sinks and exports at Palmyra, and that changes in the forest composition at the atoll scale will have an impact on carbon stocks, with decreased carbon in wood, but increased carbon in soil, resulting in a net increase in carbon storage under planned rainforest management activities. Carbon content was in the range of expected values for soils in similar environments but was unique for aboveground carbon.

It is challenging to directly compare the estimated soil values from this study to other studies due to the range of data collection techniques, spatial resolutions, depth profiles, and specific methods used to quantify SOC elsewhere. When comparing the soil carbon values to mean soil carbon values from a global dataset [43] summarized by ecoregions in the Eastern Indo-Pacific [44], we find that soil carbon content at Palmyra falls within, although closer to the lower end of the range of values found in similar environments (69.6–334.4 Mg C ha$^{-1}$). However, these global datasets are based on a larger soil depth range (1 meter), so differences based on soil depth alone would not be surprising, yet also may reflect the unique soil formation processes of atoll environments [45]. There are few comparable studies in similar atoll environments. Organic carbon percentages in Palmyra's soils are in line with expected values of 2–20% documented in the literature [45], which looked at soil characteristics across a number of Pacific atolls; however, because these values do not have associated bulk density estimates they cannot be converted to total tonnage as we did at Palmyra.

These new data echo previous research at Palmyra which found higher SOC in areas of low *C. nucifera* density [5]. *C. nucifera* stands at Palmyra may have higher SOC than *C. nucifera* monocultures in other locales. For example, in Kerala, India, samples taken at the same depth profile (0–20 cm) showed a carbon stock of 24.81 Mg C ha$^{-1}$. Although this is much lower than the estimates of soil carbon in *C. nucifera* stands at Palmyra (49.8 Mg C ha$^{-1}$), this may be due to the fact that there are active *C. nucifera* cultivation practices taking place in Kerala, where *C.*

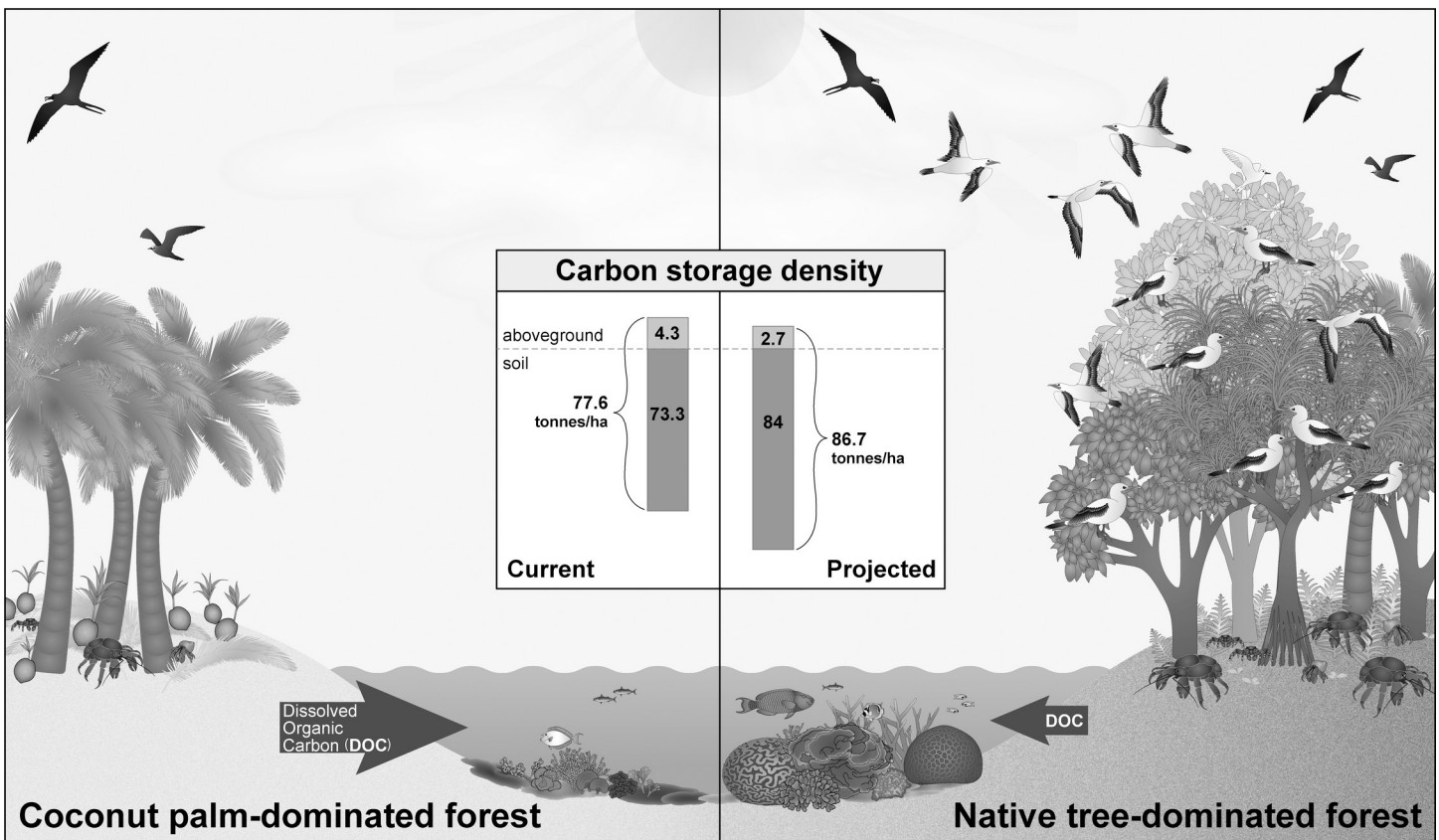

**Fig 5. Conceptual illustration of study results and other anticipated transformation and realignment benefits (bars not to scale).** The panel on the left shows a forest dominated by *C. nucifera* (current pre- transformation scenario on Palmyra), while the panel on the right shows a forest dominated by native species, following anticipated transformation and realignment. In the pre-transformation scenario, a homogenous, non-native vegetation community consisting of *C. nucifera* is associated with higher aboveground carbon values and lower soil carbon values (bars not to scale). In this scenario, palm fronds and coconut dominate the ground cover, leading to increased dissolved oxygen concentrations negatively impacting the health and diversity of nearby coral reefs. In the post-transformation scenario, a transformed, native, mixed-species vegetation community is associated with lower aboveground carbon values, with higher soil carbon values, with a total net increase in carbon storage. In this scenario, seabird and understory diversity also increases, and lower DOC concentrations mean increased coral reef health and biodiversity (Fig 5 credit: Adi Khen).

*nucifera* is strictly a monoculture. At Palmyra, *C. nucifera* can be found in mixed-species stands and plant litter (coconuts and fronds) are not removed. In general, our findings are consistent with previous studies documenting reduction in soil carbon following the conversion of native forests to agricultural plantations, inclusive of *C. nucifera* [e.g., 5, 46]. Furthermore, the carbon accumulation rate at Palmyra was consistent with the 1.3 Mg C ha$^{-1}$ yr$^{-1}$ seen during 20 years of tropical forest reforestation [47] and values of up to 5 Mg C ha$^{-1}$ yr$^{-1}$ recorded in other moist tropical environments [48].

The amount of carbon retained in soils is a function of both litter quality and quantity [49, 50]. C:N ratios and lignin contents are typically important markers of litter quality and Young et al. [51] found foliar C:N ratios are substantially higher for *C. nucifera* (32.83 ± 0.57) than for three native forest species on Palmyra (*S. sericea*, 16.15 ± 0.59; *P. grandis*, 10.92 ± 0.57; *H. foertherianum*, 9.91 ± 0.60). The C:N ratios of soils under native tree species may also be lowered by guano inputs from seabirds, which preferentially roost and nest in native forest canopies [14]. *C. nucifera* also has a relatively high lignin content (39%) [52] while recorded measurements of lignin content in other dominant tree species are lower (*P. grandis* just under 10%, *H. tiliaceus* ~15%, *P. tectorius* ~20%, and *H. foertherianum* ~13%) [53, 54]. This information suggests that litter quality is lower under *C. nucifera* canopies, which may impact SOC

accumulation rates, though long-term SOC development in some cases is more a function of litter quantity than litter quality [55]. However, at Palmyra *C. nucifera* forests have three times the litter accumulation, and yet only one third the SOC compared to native forests [5]. This suggest that enhanced litter quantities under *C. nucifera* canopies does not lead to greater SOC stabilization in this case. The balance between carbon inputs from litter fall and retention measured as SOC, as well as nearshore DOC measurements, all suggest that more carbon is leached from *C. nucifera* forests than from native forests. This might be, in part, due to the higher pH in soils from *C. nucifera* forests [5] given that high ph causes temperate soils to leach more DOC into stream and lakewater [56]; however, it is likely also due to differences in litter quality.

While estimates of SOC are generally in line with those from comparable geographies, estimates of aboveground carbon at Palmyra were much lower than the 17–255 Mg C ha$^{-1}$ reported from other tropical forests [57]. In Hawaii, for instance, mean aboveground carbon storage values were 60–130 Mg C ha$^{-1}$, with, as at Palmyra, non-native plants holding more aboveground carbon than young native plants [58]. In the tropics, aboveground carbon stocks typically exceed SOC [59, 60]; however, this pattern was not observed in this study. The relatively low aboveground biomass at Palmyra may be due to shorter tree heights (e.g., *S. sericea*) and relatively low wood density values in dominant vegetation, especially *P. grandis* and *P. tectorius*, which have softer wood texture than, for example, *H. tiliaceus*.

Although some studies tout the carbon storage benefits of agroforestry projects [61, 62], other studies show that variation among individual tree species can impact carbon storage potential in tropical ecosystems [18], and changes from native to non-native vegetation or cultivated environments can reduce carbon stocks [e.g., 62–65]. The results of this study align with the latter studies' results and further emphasize that the carbon storage outcomes of conversion will hinge on the biophysical characteristics of both the individual tree species as well as the vegetation community impacts on soil characteristics.

The increased DOC in nearshore water near *C. nucifera* forests suggests that *C. nucifera* removal could have indirect effects on nearshore communities. DOC released by macroalgae is important in the context of coral reef degradation as it contributes to coral mortality by promoting bacterial metabolism on the coral surface. DOC can be important in oligotrophic systems like Palmyra; its primary effect is to increase bacterial metabolism, which can then be a resource for some invertebrates like sponges [66], but also cause disease in corals [23]. However, given the limited sampling effort for DOC in this study, and the lack of concurrent data on nitrogen, the hypothesized net effects of terrestrial runoff on the nearshore systems at Palmyra Atoll remain speculative and in need of further study. We note that the paucity of DOC exported from native forests is opposite to the pattern seen for nitrogen associated with seabird guano from native forests. Nitrogen likely plays a different role in this system than DOC. For instance, it might increase both coral growth and phytoplankton productivity, which can support zooplankton populations [67]. Higher zooplankton abundance adjacent to native forest at Palmyra attracts manta rays, and likely has other food-web effects [14].

Stocks and flows of soil carbon at Palmyra are likely to change over time scales and climate regimes that extend beyond the scope of this study. Our calculations of accumulation rates factor in soil carbon degradation, averaged over time since the initial construction of man-made islands. Future efforts could improve on our models by more accurately and explicitly incorporating the drivers of soil carbon degradation at Palmyra.

The remote location and limited research facilities at Palmyra constrained sample sizes and restricted field techniques. Incorporating estimates of belowground biomass (e.g., roots), leaf litter, and other sources of biomass such as coconuts at Palmyra would enhance comparisons to other studies. The rapid assessment of carbon based on limited field measurements did not

sample belowground biomass, thereby potentially underestimating the amount of carbon found in tree biomass in both forest types. However, this study benefited from high resolution, remotely-sensed imagery, not typically available for such remote locations. These datasets allowed for estimates of aboveground carbon storage to be calculated at the scale of the individual tree, while simultaneously allowing for estimates of soil carbon storage potential based on community level characteristics. While the specific values estimated in this study might be refined through additional data, the key takeaways are the patterns of carbon fluctuations observed.

*C. nucifera* presence in, and in many instances dominance of, tropical oceanic island forests, through cultivation (copra) or naturalization (drift dispersal) [7] is widespread. An unreported yet likely significant portion of the world's 439 atolls [68] have experienced loss of native forest habitat through the agricultural introduction of *C. nucifera* [69]. *C. nucifera's* role in human migration and settlement throughout Oceania is notable [70], and control of *C. nucifera* to transform native forest should be balanced with the societal value provided by *C. nucifera* to Pacific Island communities. While the biocultural implications of controlling *C. nucifera* on oceanic islands have not been widely studied [71], the benefits of a landscape dominated by native vegetation facilitating seabirds, and seabird-transported nutrients are becoming more clear. For atoll ecosystems facing near-term challenges from global (climate-related) and localized impacts [72], transformation of forests from *C. nucifera* dominance to native tree dominance could not only enhance ecosystem integrity [3, 14, 73, 74] and reduce disease risk for corals [23], but also act as a natural climate solution through enhanced carbon banking [75]. This study provides further incentive to consider the climate resilience and mitigation benefits associated with intact atoll ecosystems, and specifically with forest communities beneficial to seabirds, when planning management activities.

## Supporting information

**S1 Table. List of soil samples and data used in random forest model.** Community type was assigned based on the value of the overlapping raster cell. Soil carbon percentages for each sample were binned as high (>20%), medium (10–20%), and low (<10%). Dry bulk density was assigned using the mean values collected in the field.
(DOCX)

**S2 Table. Basic wood density samples.** Samples with the same sample ID were sampled from the same individual tree, with each sample (distinguished by sample number) designated individual discs measured as replicates.
(DOCX)

**S3 Table. Summary of quadratic mean of tree diameters for each species used in analysis with the number of trees sampled.**
(DOCX)

**S4 Table. Table depicting results of health surveys for *H. foertherianum* seedlings.** Values used to determine survival rate.
(DOCX)

**S5 Table. Tables of pre and post transformation carbon values by islet.** Islet areas are included.
(DOCX)

**S6 Table. Tables of pre and post transformation carbon values by major tree species.** Note that the total values will not match those in S5 Table as values were only calculated for the

major tree species reported in Table 3.
(DOCX)

**S1 Fig. Basic wood density box plot.** Box plot used to evaluate the accuracy of corrected BWD estimates. Our corrected BWD estimates for Pisonia are significantly higher than uncorrected estimates (t-test, p = 0.037) and are indistinguishable from Pisonia BWD estimated outside Palmyra Atoll (t-test, p = 0.12). The BWD estimates from the literature (green), uncorrected Palmyra Atoll (purple) and corrected Palmyra Atoll (red) are shown in the boxplot for each genera of Palmyra Atoll trees.
(PDF)

**S1 File. R code used to generate box plot in S1 Fig.**
(R)

**S2 File. IICA subset.** Table of basic wood density (BWD) values for tropical tree species from ICCA. (Table used in R code found in S1 File).
(CSV)

**S3 File. Palmyra measurements.** Measurements of BWD from Palmyra Atoll samples. (Table used in R code found in S1 File).
(CSV)

**S4 File. Wood density values.** BWD measurements of Palmyra Atoll tree species and congeners from other sources. (Table used in R code found in S1 File).
(CSV)

## Acknowledgments

On site assistance was provided by Stefan Kropidlowski (USFWS), Nick Holmes (TNC California's Island Resilience Program), and TNC staff operating the TNC research facility at Palmyra. 2016 tree survey and soil sampling was assisted by Ana Miller-ter Kuile, Michelle Lee (NSF REU), Taylor Bogar (NSF REU), and Hillary Young (PI) of the University of California, Santa Barbara. Elisa Halewood provided the DOC protocol and analyzed the samples. Adi Khen developed the scientific illustration included in this manuscript. Any use of trade, firm, or product names is for descriptive purposes only and does not imply endorsement by the U.S. Government.

## Author Contributions

**Conceptualization:** John P. McLaughlin.

**Data curation:** Kate Longley-Wood, Mary Engels, Kevin D. Lafferty, John P. McLaughlin.

**Formal analysis:** Kate Longley-Wood, Mary Engels, Kevin D. Lafferty, John P. McLaughlin.

**Funding acquisition:** Alex Wegmann.

**Investigation:** Kate Longley-Wood, Mary Engels, Kevin D. Lafferty, John P. McLaughlin.

**Methodology:** Kate Longley-Wood, Mary Engels, Kevin D. Lafferty, John P. McLaughlin.

**Project administration:** Kate Longley-Wood, Alex Wegmann.

**Resources:** Alex Wegmann.

**Supervision:** Alex Wegmann.

**Validation:** Mary Engels, John P. McLaughlin.

**Visualization:** Kate Longley-Wood.

**Writing – original draft:** Kate Longley-Wood.

**Writing – review & editing:** Mary Engels, Kevin D. Lafferty, John P. McLaughlin, Alex Wegmann.

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
