## [Decision Letter · Decision Letter 0]

20 Aug 2021

PONE-D-21-19703

Transforming Palmyra Atoll’s forest to native-tree dominance is predicted to reduce above ground biomass and carbon export to the ocean, yet increase net carbon storage by elevating soil carbon

PLOS ONE

Dear Dr. Longley-Wood,

Thank you for submitting your manuscript to PLOS ONE. After careful consideration, we feel that it has merit but does not fully meet PLOS ONE’s publication criteria as it currently stands. Therefore, we invite you to submit a revised version of the manuscript that addresses the points raised during the review process.

Both reviewers agreed the study is relevant but consider the manuscript is partly technically sound and needs to be greatly improved. One fundamental issue is the choice of the allometric equation that should be better justified. Since the changes required are too substantive, I am willing to consider a revised version for publication in this journal, assuming you modify the manuscript according to all recommendations.

We look forward to receiving your revised manuscript.

Kind regards,

Angelina Martínez-Yrízar, Ph.D.

Academic Editor

PLOS ONE

“This project was made possible with funding from the Wildlife Conservation Society Climate Adaptation Fund—established by a grant from the Doris Duke Charitable Foundation and a grant from the National Science Foundation.”

“This project was made possible with funding from the Wildlife Conservation Society Climate Adaptation Fund (https://www.wcsclimateadaptationfund.org/)—established by a grant from the Doris Duke Charitable Foundation and a grant from the National Science Foundation.  Funding received by AW in 2018 (no grant number provided). The funders had no role in study design, data collection and analysis, decision to publish, or preparation of the manuscript.”

5. We note that Figure 1, 3 and 4 in your submission contain map images which may be copyrighted. All PLOS content is published under the Creative Commons Attribution License (CC BY 4.0), which means that the manuscript, images, and Supporting Information files will be freely available online, and any third party is permitted to access, download, copy, distribute, and use these materials in any way, even commercially, with proper attribution. For these reasons, we cannot publish previously copyrighted maps or satellite images created using proprietary data, such as Google software (Google Maps, Street View, and Earth). For more information, see our copyright guidelines: http://journals.plos.org/plosone/s/licenses-and-copyright.

Reviewers' comments:

Reviewer's Responses to Questions

**Comments to the Author**

1. Is the manuscript technically sound, and do the data support the conclusions?

Reviewer #1: Partly

Reviewer #2: Yes

2. Has the statistical analysis been performed appropriately and rigorously? 

Reviewer #1: Yes

Reviewer #2: N/A

3. Have the authors made all data underlying the findings in their manuscript fully available?

Reviewer #1: No

Reviewer #2: Yes

4. Is the manuscript presented in an intelligible fashion and written in standard English?

Reviewer #1: Yes

Reviewer #2: Yes

5. Review Comments to the Author

Reviewer #1: Although, the study has a very nice idea, ground sampling has not been done properly, which is a weakness of the paper, however, authors have tried to justify this aspect. Please see comments in the attached file.

Reviewer #2: This paper addresses the functional ecosystem-level consequences of non-native species dominance by focusing on the widely spread tropical species Cocos nucifera in the Pacific Ocean atoll of Palmyra. It quantifies C stocks in aboveground biomass and soil to establish transformation scenarios for native forest re-establishment. There are however important issues that need attention to make this study publishable. The manuscript lacks appropriate description of several key methodological procedures, better ordering in the description of results and needs more depth and insights into the mechanistic or process-based explanations of the results.

General comments

The status of C. nucifera as a non-native species is confusing, since the authors acknowledge an unclear situation (line 65), but then decide it is non-native and that the mixed forest should be the transformation target. This is a key issue that would need better explanation. As it stands, the authors’ position appears arbitrary. Estimation of aboveground biomass and C is also key for the study, so their choice of the allometric equation should be better justified (see also specific comments). It would be very important to know if the equation applies equally well to trees and palms. Also, the logic behind the modeling needs better description, especially for a non-specialist in remote sensing technology and its applications. It seems awkward to me to start the results by describing the projections rather than the actual observations. The discussion in general needs major revision in terms of order and paragraph structure. For example, the opening paragraph highlights some results, but then goes into a list of caveats for the study. In my opinion, these should be considered after discussion of the major contributions of the research. Also, paragraph structure needs revision; several paragraphs introduce an issue and end up with a different one without closure of the initial arguments. There are also conceptual issues that need better treatment. For example, soil C accumulation is a key variable for the study, but no explanation or allusion to decomposition processes, variation in litter C quality among C. nucifera and other species and no mechanistic explanation is included in the explaining of differences in soil C accumulation between vegetation types. This is crucial since the transformation projection highlights especially the role of changes in soil C. Also, the processes involved in increased DOC fluxes to the ocean from C. nucifera vegetation need a better discussion (see below).

Specific comments

line 65 - It’s not clear if C. nucifera is non-native or not, but the authors then decide it is not. Please make sure these arguments are clear.

lines 139-140 - Was this procedure followed for every soil collection? Please clarify.

lines 148-161 - It is not clear why you start by describing th modeling of C distribution before what you did with the actual measurements. Also, the logic behind the modeling approach should be better explained by providing a bit more contexta. It seems also akward that the modeling is described before the procedures to estimate aboveground biomass and C.

line 185 - please avoid the use of the term sequestration; the term accumulation is the correct one.

line 205 - why use Chave et al. (2005) and not the more recent Chave et al. (2014)?

line 206 - BWD is missing from the equation; is this an appropriate equation for palms? More explanation and justification is needed since aboground biomass is a key variable of the study.

lines 207-212 - this information on BWD is mentioned previously in the manuscript, there is no need to repeat here.

line 268 - it seems awkward to me to start the description of results with the random forest analyses rather than with the actual measurements. I think it needs re-organization.

line 299 - how do the authors arrive at 848.2 Mg C? Please explain.

lines 301-306 - this is not needed here, it´s already in the methods.

line 309 - the table indicates 33.4 not 33.3

line 335 - perhaps include the value for C. nucifera here?

lines 367-368 - I don’t think that measurements of BWD for unrecorded species should be included as a highlight of the study.

lines 379-395 - this paragraph needs better focus; it mixes different issues.

line 383 - the authors should provide here data for similar environments and references.

lines 384-385 - this sentence seems to contradict the previous one.

lines 408-411 - what about decomposition rates and C quality? no mention on litter permanence on the soil and no explanation on the role of pH for DOC transport. The processes need to be discussed.

lines 412-418 - Is it possible that aboveground C estimates are influenced by the equation used?

line 414 - these values are not rates.

line 415 - the sentence on ABC and SOC is out of place here.

lines 426-437 - a better discussion of the issues is needed here. Why expect that DOC would have similar effects than N fluxes? More insights and depth are needed.

line 432 - what hypothesis they refer to?

lines 438-444 - this paragraph seems misplaced here.

lines 445-457 - this paragraph reads more like introduction than a conclusive paragraph, except at the end. What would the resulting benefits from soil C enhancement under native tree dominance in atolls represent with the perspective of sea level rise? How is ocean acidifcation related to the benefits of native tree dominance? A better discussion of these issues should constitute the ending paragraph for the manuscript.

6. PLOS authors have the option to publish the peer review history of their article (what does this mean?). If published, this will include your full peer review and any attached files.

Reviewer #1: **Yes: **RK Chaturvedi

Reviewer #2: No

---

## [Author Response · Author response to Decision Letter 0]

15 Dec 2021

Dear PLOS One Editorial Staff,

On behalf of my co-authors, I would like to kindly thank the reviewers and editorial staff for their time in reviewing the manuscript submission PONE-D-21-19703 entitled “Transforming Palmyra Atoll’s forest to native-tree dominance is predicted to reduce above ground biomass and carbon export to the ocean, yet increase net carbon storage by elevating soil carbon.” The feedback and guidance provided has greatly strengthened the manuscript. 

We have incorporated reviewer comments and revised our estimates of aboveground carbon storage. First, we revised our basic wood density (BWD) estimates according to reviewer recommendations. BWD estimates for Palmyra trees are now comparable to BWD estimates for congeners measured elsewhere. These corrected BWD values informed our revised estimates of aboveground density, which used the Chave 2014 allometric equation for aboveground biomass suggested by the reviewers. 

While they did not alter our conclusions, these revisions improved the accuracy or our results and we appreciate the reviewers help in making them. We have added Dr. John McLaughlin as a coauthor and submitted a change of authorship form. John provided some of the original Palmyra data in the manuscript and his contributions to revising our BWD estimates rose to the level of coauthorship. We have also changed the title of the manuscript to better communicate the primary message of the paper.

Other changes to the manuscript are addressed point-by-point in the following pages. We look forward to hearing from you regarding our submission, and we would be glad to respond to any additional questions or comments. 

Responses to editorial comments to ensure alignment with journal requirements

• In response to the editor’s request, we adjusted manuscript headings to match styles in formatting guidance document (still need to do tables and figures).

• We have removed the funding statement from the Acknowledgement section. The new funding statement should read: “This project was made possible with funding from the Wildlife Conservation Society Climate Adaptation Fund – established by a grant from the Doris Duke Charitable Foundation and a grant from the National Science Foundation. Additional data collection was funded by the NSF DEB #1457371.”

• We have updated our data availability statement. Since the original manuscript submission, we have submitted the spatial data associated with this project (wood, soil, and DOC sampling locations and transformation area boundaries) to the Environmental Data Initiative Data Portal, a publicly-accessible online repository. The data will be published prior to the publication of this manuscript, and the doi numbers can be provided following manuscript acceptance. Between these data, and the data provided in the extensive supplementary materials, we believe that this addresses the reviewer comment regarding data accessibility, but if the editorial staff believes that another key component is missing, please contact us and we will address that. 

• In response to the editorial comment, we’d like to confirm that ethics statement is now found in the Methods section only in the revised version of the manuscript.

• In response to the inquiry about potentially copyrighted material in Figures 1, 3, and 4, we note the following information: In Figure 1, the island basemap in main panel uses a shapefile available from the USGS Sciencebase.gov data portal, which is in the public domain and not copyrighted. In the global inset map in Figure 1, we have changed the basemap so that it uses data from Natural Earth, a public domain site. Similarly, Figure 4 uses the same source as Figure 1 for its basemap. In the revised version of the manuscript, we note the source of these basemaps in the figure caption. In both Figures 1 & 4, other data displayed on the map were generated by authors in this study. Figure 3 only displays data generated from this study (in other words, there are no basemaps, only the maps generated through the analysis described in the methodology, where source datasets are cited). Based on this information, we do not believe that there are any copyright concerns. 

• As above, we note the addition of a co-author to this paper and a change of authorship form has been submitted electronically. 

Responses to specific manuscript content comments, by line

We thank the reviewers for the opportunity to clarify and improve our methods. In the following section, reviewer comments are indicated in bold, with the author response following in normal text. For small editorial suggestions we acknowledge that the change has been accepted. For more detailed comments, we provide more detail on either changes made to the manuscript, or, in the few instances where appropriate, a justification for keeping the text as-is. 

Line 5: Replace cycle with dynamics. Change accepted. 

Line 33: Replace cycle with dynamics. Change accepted. 

Line 38: Remove the word “the”. Change accepted

Line 38: Replace “predicted” with “projected”. Change accepted 

Line 39: Replace “to” with “an” and insert “in” and “thereby”. Changes accepted 

Line 57 – 59: “It would be nice to read the aim of study in the last paragraph. Also, briefly describe your hypothesis”. We agree with the reviewer comment. The text describing the aim of the study has been moved to the end of the paragraph, and a sentence stating the hypothesis has been added. 

Line 65: Replace “shrank” with “shrinked”. Change accepted. 

Line 65: It’s not clear if C. nucifera is non-native or not, but the authors then decide it is not. Please make sure these arguments are clear. We agree that this point required clarification. We have changed wording to back up our assumptions around whether C. nucifera should be considered non-native, and referenced other studies that point to it being invasive.

Line 66: Add comma. Change accepted. 

Line 69: Add “Moreover”. Change accepted. 

Line 81: Change text from “C. nucifera removal work began in 2019” to “The forest transformation programme at Palmyra started in 2019”. Change accepted. 

Line 86: Also see: Chaturvedi et al. 2011, Forest Ecology and Management 262(8): 1576-1588. Chaturvedi & Raghubanshi 2015, Forest Ecology and Management 339(2015): 11–21. We thank the reviewer for supplying this additional reference and have added this to the manuscript. 

Line 90: Suggestion to provide the quantitative data of precipitation, pH, and decomposition rate in a supplementary table for better information about the study sites. We would like to clarify that these data do not exist at the scale of the individual sampling sites, and this information was intended to provide information about general conditions at Palmyra. The text was altered slightly to clarify this point. However, we did add both to the methods and to the results information about air temperature and shade at each DOC sampling site. 

Line 90: Suggestion to shift sentence to last sentence of the paragraph as a concluder sentence. Suggestion accepted. 

Line 91: Add “Besides”. Change accepted. 

Line 94: Add comma. This paragraph has been reordered to address previous comments, rendering this change unnecessary. 

Line 130: Comment: “For soil organic carbon, the supplementary table only shows categorical data in the form of high, medium and low organic carbon percentage. The quantitative data can be seen in figures, but only in the form of maps, and for dominant species communities. In methods, the sampling density for soil organic carbon has also not been clearly described. At present, we can only understand that at most only one sample has been collected from each islet. As we know that soil organic carbon is quite heterogeneous and sampling density highly influence the results. I will suggest to clearly describe the sampling design for more transparency.” 

 In order to address this comment, we made the following changes: We have included the calculated carbon percentages in Supplementary Table 1 to address the reviewer comment regarding a better linkage between the quantitative data values and the categorical high, medium, and low bins. While Supplementary Table 1 as originally submitted contained information on the number the samples per islet, we have added a summary table (now Table 2) to the body of the manuscript to summarize the number of soil samples extracted from each islet. We have also modified the text to indicate that the bins for soil carbon concentration were applied to account for known heterogeneity. We did this as we know that individual samples may not be representative, so we used a conservative approach to account for this heterogeneity. We have added additional text to explain the impact of sampling design on the final sampling density. 

lines 139-140 - Was this procedure followed for every soil collection? Please clarify. This procedure was followed for every soil sample. We have clarified this in the text. 

lines 148-161 - It is not clear why you start by describing the modeling of C distribution before what you did with the actual measurements. Also, the logic behind the modeling approach should be better explained by providing a bit more context. It seems also awkward that the modeling is described before the procedures to estimate aboveground biomass and C. 

We addressed the first comment by moving the description of binning the carbon estimates to the beginning of the paragraph. To the reviewers second point, we experimented with different ways of organizing this paper. Earlier versions of the paper lumped described field data collection for above and belowground carbon in one section, followed by modelling approaches. Ultimately, we elected to describe the entire methodology for soil carbon before describing the methodology for describing aboveground biomass, as the procedures were so different and doing so improved the overall flow of the paper. 

Line 185 - please avoid the use of the term sequestration; the term accumulation is the correct one. We have replaced the term “sequestration” with “accumulation” throughout the manuscript. 

line 205 - why use Chave et al. (2005) and not the more recent Chave et al. (2014)? The calculations were updated using the Chave et al. (2014) allometric equation. 

line 206 - BWD is missing from the equation; is this an appropriate equation for palms? More explanation and justification is needed since aboveground biomass is a key variable of the study. 

In the original equation, BWD was present in the equation and was represented by the letter 𝜌, as described in the sentence immediately preceding the equation. This same symbol is used in the new equation from Chave et al. (2014) that was used to recalculate the aboveground biomass values as suggested by Reviewer 2. The Chave et al. (2014) article notes that this equation is appropriate across pantropical environments and was based on field measurements taken at 58 sites, globally. Given the broad geographical distribution of C. nucifera across pantropical environments, and the absence of a site-specific allometric equation for C. nucifera, we believe this equation is appropriate for these estimates. We have added text to justify this choice. 

lines 207-212 - this information on BWD is mentioned previously in the manuscript, there is no need to repeat here. The repeated information has been removed. 

Line 222: “I will suggest to take the actual land area, and not the areas covered by the canopy. The reason is the difference in canopy structure of C. nucifera as compared to other tree species. I suspect that the canopy of C. nucifera is more open, exhibiting lesser canopy width, compared to the other selected tropical trees. This could also be the reason for more aboveground carbon for C. nucifera stands as compared to stands dominated by other tropical trees.” We agree with the reviewer and this is the approach that we took in the original analysis, though we acknowledge that this may not have been fully evident in our description. To clarify, we only excluded areas where there were absolutely no trees in the vicinity (e.g., the paved runway), but we wouldn’t have excluded slivers of land due to small canopy gaps. We clarified this in the text. 

Line 250: “Numbers of samples very low; how to defend?” We acknowledge the reviewer’s point about low sample size and refer to the note in the discussion acknowledging the low sample size and the preliminary nature of the data and subsequent conclusions. We have added a point in the discussion to note that despite the small sample size, the magnitude of difference among samples is still quite large. 

line 268 - it seems awkward to me to start the description of results with the random forest analyses rather than with the actual measurements. I think it needs re-organization. We agree with the reviewer’s comment and have added more information on the measurements at the beginning of this section. To do this, we have added a summary table to this beginning of this section that provides more detail on soil carbon and dry bulk density measurements. To provide further transparency, a column providing the soil carbon concentration for each sample has been added to Supplementary Table 1. 

Line 296: Change “carbon soil” to “soil carbon”. Change accepted. 

line 299 - how do the authors arrive at 848.2 Mg C? Please explain. In the soil modelling methods section we describe the assumptions and calculations that were undertaken to complete these (lines 176 – 187 in the original manuscript); however, we have added a table that clarifies these calculations. In making this correction, it should be noted that we resolved an error in the original calculation which was corrected in the revised version of the manuscript. 

lines 301-306 - this is not needed here, it´s already in the methods. This has been removed. 

Line 301: For reliable method please see Chave et al. 2006. This paper provides an overview of calculating basic wood density which is then used in the allometric equation for aboveground biomass. This was sound advice and we have improved our BWD estimates accordingly. We corrected our BWD estimates after Vieilledent et al. (2018). This formula is nearly identical to Chave et al. (2006), but 5% more accurate. To do this, we added literature estimates of the fiber saturation point and volumetric shrinkage (Cordero 1971) for Palmyra trees. From these values we were able to derive the parameters for the Vieilledent/Chave equations. To evaluate the accuracy of these corrected BWD estimates we compared them to literature estimates for congeners outside Palmyra. Our corrected BWD estimates for Pisonia are significantly higher than uncorrected estimates (t-test, p = 0.037) and are indistinguishable from Pisonia BWD estimated outside Palmyra (t-test, p = 0.12). The BWD estimates from the literature (green), uncorrected Palmyra (purple) and corrected Palmyra (red) are shown in boxplot format below, for each genera of Palmyra trees. This box plot, as well as the R code and data tables used to produce the box plot have been included in the supplementary materials for greater transparency. 

line 309 - the table indicates 33.4 not 33.3. This value was recalculated and verified against the table. 

 line 335 - perhaps include the value for C. nucifera here? Because the values supporting this sentence are provided later in the paragraph, we deleted the sentence referred to by the reviewer to reduce confusion and redundancy. 

 lines 367-368 - I don’t think that measurements of BWD for unrecorded species should be included as a highlight of the study. This has been removed. 

 lines 379-395 - this paragraph needs better focus; it mixes different issues. We agree with the reviewer and we have moved the main headline of results up to the first paragraph of the discussion to provide clarity and focus. Then, in the subsequent paragraphs, we go on to discuss the soil carbon and aboveground carbon components in further detail. We have broken the two components out into separate paragraphs to emulate the structure of the rest of the paper to make it easier to follow.

 line 383 - the authors should provide here data for similar environments and references. Here, we refer the reviewer to subsequent paragraphs where references to support this statement are provided. 

 lines 384-385 - this sentence seems to contradict the previous one. We agree with the reviewer and in the revised version of the manuscript we have added a clarification that although they are within the range, they are on the lower end of the range. 

 lines 408-411 - what about decomposition rates and C quality? no mention on litter permanence on the soil and no explanation on the role of pH for DOC transport. The processes need to be discussed. 

A brief explanation and reference regarding the relationship between pH and DOC transport has been provided in the revised version of the manuscript. 

 lines 412-418 - Is it possible that aboveground C estimates are influenced by the equation used? 

Following comments on the first submission, we re-ran the calculations using the Chave 2014 equation rather than the one found in Chave 2005, and included corrected bwd values to account for possible underestimates [as described previously in this letter]. While we did see the estimates of aboveground carbon increase somewhat, they still did not fall within the typical range reported in the literature. 

 line 414 - these values are not rates. . This has been corrected. 

 line 415 - the sentence on ABC and SOC is out of place here. This paragraph is intended to convey the somewhat anomalous results of the aboveground carbon analysis, and so we have kept it in; however, we have added a topic sentence to better focus the discussion, and added additional clarifying text. 

lines 426-437 - a better discussion of the issues is needed here. Why expect that DOC would have similar effects than N fluxes? More insights and depth are needed. We have clarified that DOC does not have similar effects of nitrogen fluxes. 

line 432 - what hypothesis they refer to? Here we are referring to the effects of terrestrial runoff in nearshore systems at Palmyra. We have added clarifying text. 

 lines 438-444 - this paragraph seems misplaced here. We agree with this comment. We have moved the other paragraphs discussing caveats and other guidance for interpreting the results of the study. We moved text that had described the sampling limitations down to this part of the discussion so the caveats are all found within the same part of the discussion. 

lines 445-457 - this paragraph reads more like introduction than a conclusive paragraph, except at the end. What would the resulting benefits from soil C enhancement under native tree dominance in atolls represent with the perspective of sea level rise? How is ocean acidifcation related to the benefits of native tree dominance? A better discussion of these issues should constitute the ending paragraph for the manuscript. We agree that some of the information provided here regarding the global issues of C. nucifera spread are needed in the introduction, so we added text to the effect at the beginning of the paper. However, we believe that revisiting this topic in the discussion is also valuable to set the results in global context, and so this information has been retained in this concluding paragraph. We have also refocused the discussion of global impacts somewhat to focus on the fact that carbon storage is a co-benefit to other expected atoll-level benefits such as seabird habitat and coral disease risk reduction. 

Line 456: add “ing”. The structure of this sentence was changed so this comment is no longer applicable.

---

## [Editor Report · Decision Letter 1]

31 Dec 2021

Transforming Palmyra Atoll to native-tree dominance will increase net carbon storage and reduce dissolved organic carbon reef runoff

PONE-D-21-19703R1

Dear Dr. Longley-Wood,

We’re pleased to inform you that your manuscript has been judged scientifically suitable for publication and will be formally accepted for publication once it meets all outstanding technical requirements.

Kind regards,

Angelina Martínez-Yrízar, Ph.D.

Academic Editor

PLOS ONE

Additional Editor Comments (optional):

The ms. was greatly improved in clarity. However, there are still some minor errors that need your attention.

L30, 90, 116 and 188, replace the word “sequestration” with “accumulation” as suggested by the reviewer

L134 (Table 2, third row), other tables and throughout the text, correct the species name ….foerthenarium….

L198, for clarity replace (BWD) with (BWD; i.e., the ratio of dry mass over green volume)

L218, insert BWD in brackets ….basic wood density (BWD) values by species….

L216, explain what do you mean by “corrected BWD values"

Table 2, insert a space in species names; i.e., P.grandis to P. grandis. Check this error throughout tables and text (L451, L455, etc.)

Table 3, correct the species name …T. cattapa.... Also in L318

Table 4, make OC% column wider to make clear the correspondence between OC% ranges and bins

L426, delete “soil carbon” before the word dataset

L473, include in brackets the WD values of each of the three species mentioned here

L746, for clarity replace …Table of BWD values… with …Table of basic wood density (BWD) values…

Although all captions of Supporting information are provided in the manuscript file, also include each caption in each of the supporting information file itself

Supplementary Figure legends, put the whole name “Palmyra Atoll” when referring to the Palmyra forest, site, or samples

Check for consistency of References style (Journal name abbreviations, spaces, punctuation) according to Plos One guidelines
---

## [Editor Report · Acceptance letter]

11 Jan 2022

PONE-D-21-19703R1 

Transforming Palmyra Atoll to native-tree dominance will increase net carbon storage and reduce dissolved organic carbon reef runoff 

Dear Dr. Longley-Wood:

I'm pleased to inform you that your manuscript has been deemed suitable for publication in PLOS ONE. Congratulations! Your manuscript is now with our production department. 

Kind regards, 

on behalf of

Dr. Angelina Martínez-Yrízar 

Academic Editor

PLOS ONE